# RECALL LOSS FOR IMBALANCED IMAGE CLASSIFICATION AND SEMANTIC SEGMENTATION

## ABSTRACT

Class imbalance is a fundamental problem in computer vision applications such as semantic segmentation and image classification. Specifically, uneven class distributions in a training dataset often result in unsatisfactory performance on under-represented classes. Many works have proposed to weigh the standard cross entropy loss function with pre-computed weights based on class statistics such as the number of samples and class margins. There are two major drawbacks to these methods: 1) constantly up-weighing minority classes can introduce excessive false positives especially in semantic segmentation; 2) many recent works discovered that pre-computed weights have adversarial effects on representation learning. In this regard, we propose a *hard-class mining loss* by reshaping the vanilla cross entropy loss such that it weights the loss for each class dynamically based on changing *recall* performance. We show mathematically that the novel recall loss changes gradually between the standard cross entropy loss and the well-known inverse frequency cross entropy loss and balances precision and accuracy. We first demonstrate that the proposed loss effectively balances precision and accuracy on semantic segmentation datasets, and leads to significant performance improvement over other state-of-the-art loss functions used in semantic segmentation, especially on shallow networks. On image classification, we design a simple two-head training strategy to show that the novel loss function improves representation learning on imbalanced datasets. We outperform the previously best performing method by 5.7% on Place365-LT and by 1.1% on iNaturalist.

## 1 INTRODUCTION

Dataset imbalance is an important problem for many computer vision tasks such as semantic segmentation and image classification. In semantic segmentation, imbalance occurs as a result of natural occurrence and varying sizes of different classes. For example, in an outdoor driving segmentation dataset, light poles and pedestrians are considered minority classes compared to large classes such as building, sky, and road. These minority classes are often more important than large classes for safety reasons. In image classification, imbalance can occur as a result of data collection. Some classes are more difficult to obtain data for than others. For example, the inaturalist dataset (Van Horn et al., 2018) has collected images of over 8000 natural species. Since some species are rare, the dataset exhibits the notorious long-tail distribution. When presented with imbalanced datasets, the standard cross entropy loss often yields unsatisfactory results as the training process naturally biases towards large classes resulting in low accuracy and precision on small classes.

Researchers have studied the imbalance problem for classification, detection, and segmentation extensively. Most prior research has been on designing balanced loss functions. We classify existing loss functions under three categories: *region-based losses*, *statistics-balanced losses* and *performance-balanced losses*. **Region-based losses** directly optimize region metrics (e.g., Jaccard index (Rahman & Wang, 2016)) and are mainly popular in medical segmentation applications; **Statistics-balanced losses** (e.g., LDAM (Cao et al., 2019), Class-Balanced (CB) loss (Cui et al., 2019)) up/down weighs the contribution of a class based on its class margin or class size; however, they tend to encourage excessive false positives in minority classes to improve mean accuracy especially in segmentation. A recent study in Zhou et al. (2020) also shows that the weighting undermines the generic representation learning capability of the feature extractors; **Performance-balanced losses** (e.g., focal loss (Lin et al., 2017)) use a certain performance indicator to weigh

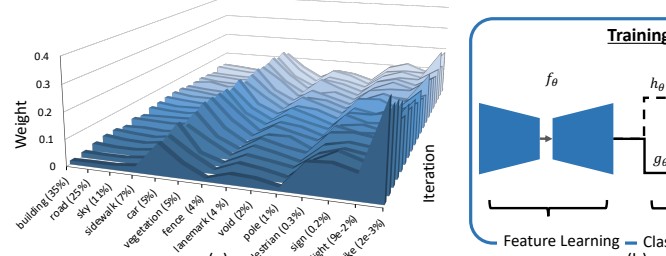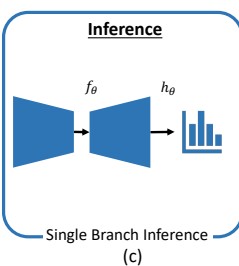

(a)  (b)  (c)

Figure 1: **(a)** We show normalized recall weights over iterations for recall loss. The horizontal axis is arranged in descending order in terms of pixel percentage. % indicates class pixel percentage. The recall weights change dynamically according to the performance metric recall. **(b)** We design a Simple Decoupled Network (SDN) to decouple representation and classifier learning. **(c)** In inference, only one branch is used in SDN.

the loss of each class. As an example, focal loss assumes that the difficulty of a class is correlated with imbalance and can be reflected by the predicted confidence. However, it has not been very successful in other applications for dealing with imbalance as reported by Cui et al. (2019). We investigate the reasons of failure in Appendix A.1. Besides various losses, another thread focuses on training strategies to decouple classifier and representation learning in image classification such as the two-stage (Kang et al., 2020) and two-branch (Zhou et al., 2020) approaches. The decoupling approaches have shown state-of-the-art performance compared to other carefully designed losses. As studied by Zhou et al. (2020), statistics-balanced losses might even negatively affect representation learning because they *always* upweigh a minority class and ignores many more examples from the large classes.

We propose a novel performance-balanced loss using the *recall* metric to address the imbalance problem. The recall loss down/up weighs a class based on the training *recall* performance of that class. It is an example of *hard class mining* as supposed to the hard example mining strategy in the focal loss. Unlike the statistics-balanced losses, the recall loss dynamically changes its weights with training based on per-class recall performance (see fig. 1(a)). The dynamism is the key to overcome many drawbacks of the statistics-balanced losses. In our experiments, the CB loss improves accuracy at the expense of Intersection over Union (IOU) which considers false positives in semantic segmentation. However, our recall loss can effectively balance between precision and recall of each class, and hence, it improves accuracy but maintains a competitive IOU. Experiments on two benchmark semantic segmentation datasets demonstrate that the proposed recall loss shows significantly better performance than state-of-the-art loss functions used in prior works. We also show that while statistics-balanced losses negatively affect representation learning, the recall loss improves representation learning for imbalanced image classification and achieves state-of-the-art results with our simple decoupled network (fig. 1(b),(c)) on two common benchmarks. Specifically, we outperform previous state-of-the-art methods on Place-LT by 5.7% and iNaturalist2018 by 1.1%.

Our main contributions are summarized below.

- We introduce a novel loss function based on the metric recall. Recall loss weighs the standard cross entropy loss for each class with its instantaneous training recall performance.

- The proposed recall loss learns a better semantic segmentation model that provides improved and balanced performance of accuracy and IOU. We demonstrate the loss on both synthetic and real semantic segmentation datasets.

- The proposed loss also improves feature learning in image classification. We show state-of-the-art results on two common classification benchmarks with a simple decoupled network.

## 2 RELATED WORK

**Imbalance in Image Classification.** Various losses have been proposed to deal with imbalance or long-tail distributions in image classification. Cost-sensitive loss (Khan et al., 2017) proposes to

iteratively optimize both the model parameters and also a cost-sensitive layer which is integrated into the cost function (more in Appendix B). Lifted Loss (Oh Song et al., 2016) considers all positive and negative pairs in a mini-batch. Range loss (Zhang et al., 2017) pushes examples in the same class together while forcing different class centers away from each other. More complicated margin-based approaches, (Dong et al., 2018; Khan et al., 2019; Hayat et al., 2019) are discussed in the Appendix B. Class-Balanced Loss (Cui et al., 2019) motivates a weighted cross entropy loss with the concept of *effective number of samples* in each class. LDAM (Cao et al., 2019) also derives a weighted cross entropy loss based on margins between classes. However, DecoupleRC (Kang et al., 2020) pointed out that balanced losses might negatively affect the representation learning process; hence, classifier learning and representation learning should be separated. OLTR (Liu et al., 2019) first learns a good representation and uses an attention mechanism to learn a balanced classifier. In the same spirit, DRW (Cao et al., 2019) uses a two-stage training, and BBN (Zhou et al., 2020) proposes a two-branch network with a custom training schedule. Both methods emphasize generic representation learning in the beginning and rebalancing the small classes at a later stage. However, both methods require extensive experiments for finding a good learning schedule. Drawing from the same idea, we design a Simple Decoupled Network (SDN) that uses two classification heads where one head is responsible for feature extractor training and the other for classifier training.

**Imbalance in Image Segmentation.** In image segmentation, Dice and Jaccard indices (Intersection over Union) are commonly used as the evaluation metrics. However, the most common training criterion, cross entropy, does not directly optimize these metrics. In medical imaging, researchers proposed to optimize soft/surrogate version of these indices. SoftIOU (Rahman & Wang, 2016) proposes to optimize a soft version of the Jaccard index; Lovasz Softmax (Berman et al., 2018) also optimizes the Jaccard index based on the Lovasz convex extension; SoftDice (Sudre et al., 2017) optimizes a soft version of the Dice index and similarly softTversky (Salehi et al., 2017) optimizes a soft Tversky index. Table 1 in Appendix 3.4 provides an overview of the different indices. However, concerns have been raised in Taghanaki et al. (2019) on whether these losses consider the trade-off between false positives and false negatives. We show that they also tend to yield high mean accuracy at the expense of lower mean IOU, whereas our loss improves accuracy while maintaining a competitive IOU.

**Imbalance in Object Detection.** Imbalance is also a problem in object detection where the foreground-background imbalance is extreme and undermines learning. Online Hard Example Mining (OHEM) (Shrivastava et al., 2016) proposes to find hard examples by ranking the losses and only keeping those with the highest losses. Seesaw Loss (Wang et al., 2020) proposes to dynamically weight the cross entropy loss with cumulative class ratios. Focal Loss (FL) (Lin et al., 2017) chooses to down weigh easy samples and emphasize hard samples by weighting each sample by $1-p$ where $p$ is the predicted probability for the sample. The weight for each sample dynamically changes with training and the method never completely discards any samples. Focal loss is especially successful because it is easy to implement and proves effective in the binary foreground-background imbalance setting. We compare the proposed method with these losses on image classification and semantic segmentation.

## 3 RECALL LOSS

### 3.1 MOTIVATION: FROM INVERSE FREQUENCY LOSS TO RECALL LOSS

To motivate our proposed loss, we first analyze the standard cross entropy loss. Let $\{x_n, y_n\} \forall n \in \{1, ..., N\}$, where $x_n \in R^d, y_n \in \{1, ..., C\}$ denote the set of training data and corresponding labels. Let $P_n$ denotes the predictive softmax-distribution over all classes for input $x_n$ and $P_n^i$ denotes the probability of the $i$-th class. The cross entropy loss used in multiclass classification is defined as:

$$CE = -\sum_{n=1}^{N} \log(P_n^{y_n}) = -\sum_{c=1}^{C} \sum_{n:y_n=c} \log(P_n^{y_n}) = -\sum_{c=1}^{C} N_c \log(P^c) \qquad (1)$$

where $P^c = (\prod_{n:y_n=c} P_n^{y_n})^{1/N_c}$ denotes the *geometric mean confidence* of class $c$ and $N_c$ denotes the number of samples in class $c$. As shown in Eq. 1, the conventional cross entropy optimizes the geometric mean confidence of each class weighted by the number of pixels in each class. When there is a significant class imbalance in the dataset, the loss function biases towards large classes as a result of larger $N_c$.

One commonly used loss for imbalanced datasets is *inverse frequency cross entropy* loss (Eigen & Fergus, 2015; Badrinarayanan et al., 2017) which assigns more weight to the loss of minority classes. Let $N$ denote the total number of pixels in the training set and $N_c$ denotes the number of pixels belonging to class $c \in \{1, .., C\}$. The frequency of a class is calculated as $freq(c) = N_c/N$. We show that while the unweighted cross entropy loss optimizes the overall confidence, the loss weighted by inverse frequency optimizes mean confidence. If we use an inverse frequency weighting, the loss is rebalanced. Note we leave out the $N$ in $freq(c)$ as it is shared by all classes.

$$InvCE = -\sum_{c=1}^{C} \frac{1}{freq(c)} N_c \log(P^c) = -\sum_{c=1}^{C} \frac{1}{N_c} N_c \log(P^c) = -\sum_{c=1}^{C} \log(P^c) \qquad (2)$$

As shown in Eq. 2, the weighted loss optimizes the geometric mean of accuracy directly. However, the inverse frequency loss might not be optimal in practice because it over-weighs the minority classes and introduces excessive false positives, i.e., it sacrifices precision for recall. This problem is especially severe in semantic segmentation (Chan et al., 2019). Applying the inverse frequency loss to segmentation increases recall for each class. However, the improvement comes at the cost of excessive false positives, especially for small classes.

While the inverse frequency loss solves the problem of imbalance, it focuses only on improving one aspect of the problem in classification, i.e. the recall of each class. To solve this issue, we propose to weigh the inverse frequency loss in Eq. 2 with the *false negative* ($FN_c$) counts for each class. The first insight is that the $FN_c$ is bounded by the total number of samples in a class and zero, i.e.

$$N_c \geq FN_c \geq 0 \qquad (3)$$

By weighting the inverse frequency cross entropy loss in Eq. 2 by the false negative counts for each class, we obtain a *moderate* loss function which sits between the regular cross entropy loss and inverse frequency loss. We want to note that the idea of finding a middle ground between these two loss functions has been explored in different forms. For example, the BBN (Zhou et al., 2020) method explicitly uses an adaptor function that controls the contribution of the two losses. However, an obvious drawback is that the adaptor function needs to be extensively searched based on empirical evidence and intuition.

$$RecallCE = -\sum_{c=1}^{C} FN_c \log(P^c) = -\sum_{c=1}^{C} \frac{FN_c}{N_c} N_c \log(P^c) = -\sum_{c=1}^{C} \frac{FN_c}{FN_c + TP_c} N_c \log(P^c)$$
$$(4)$$

As Eq. 4 shows, the loss can be implemented as the regular cross entropy loss weighted by class-wise *false negative rate (FNR)*. The second insight is that minority classes are most likely more difficult to classify with higher FNR and large classes with smaller FNR. Therefore, similar to inverse frequency loss, gradients of minority classes will be boosted and gradients of majority classes will be suppressed. However, unlike frequency weighting, the weighting will not be as extreme as motivated in Eq. 3. In the next section, we will derive the final dynamic form and compare it to the other performance-balanced loss: the focal loss (Lin et al., 2017).

### 3.2 MOTIVATION: FROM FOCAL LOSS TO RECALL LOSS

The previous section proposed to weigh cross entropy with the false negative rate of each class. Unlike frequency and decision margin (Cao et al., 2019) which are characteristics of the dataset, FNR is a metric of a model's performance. As we continue to update the model's parameters, FNR changes. Therefore, the weights for each class change dynamically to reflect a model's instantaneous performance. We rewrite Eq. 4 and introduce a subscript $t$ to denote the time dependency.

$$RecallCE = -\sum_{c=1}^{C} (1 - \frac{TP_{c,t}}{FN_{c,t} + TP_{c,t}}) N_c \log(p^{c,t}) = -\sum_{c=1}^{C} \sum_{n:y_i=c} (1 - \mathcal{R}_{c,t}) \log(p_{n,t}) \quad (5)$$

where $\mathcal{R}_{c,t}$ is the recall for class $c$ at optimization step $t$. $n : y_i = c$ denotes all samples such that the ground truth label $y_i$ is class $c$.

The other performance-balanced loss is focal loss (Lin et al., 2017). It is developed originally for background-foreground imbalance in object detection. The loss function weighs the cross entropy

| | $Recall(\mathcal{G}_c, \mathcal{P}_c)$ | $Precision(\mathcal{G}_c, \mathcal{P}_c)$ | $Dice(\mathcal{G}_c, \mathcal{P}_c)$ | $Jaccard(\mathcal{G}_c, \mathcal{P}_c)$ | $F1(\mathcal{G}_c, \mathcal{P}_c)$ | $Tversky(\mathcal{G}_c, \mathcal{P}_c)$ |
|---|---|---|---|---|---|---|
| Set Rep. | $\frac{|\mathcal{G}_c \cap \mathcal{P}_c|}{|\mathcal{G}_c|}$ | $\frac{|\mathcal{G}_c \cap \mathcal{P}_c|}{|\mathcal{P}_c|}$ | $\frac{2|\mathcal{G}_i \cap \mathcal{P}_c|}{|\mathcal{P}_c| + |\mathcal{G}_c|}$ | $\frac{|\mathcal{G}_c \cap \mathcal{P}_c|}{|\mathcal{G}_c \cup \mathcal{P}_c|}$ | $\frac{|\mathcal{G}_c \cap \mathcal{P}_c|}{|\mathcal{G}_c \cup \mathcal{P}_c| + \frac{1}{2}|\mathcal{P}_c| + \frac{1}{2}|\mathcal{G}_c|}$ | $\frac{|\mathcal{G}_c \cap \mathcal{P}_c|}{|\mathcal{G}_c \cup \mathcal{P}_c| + \alpha|\mathcal{P}_c| + \beta|\mathcal{G}_c|}$ |
| Boolean Rep. | $\frac{TP_c}{TP_c + FN_c}$ | $\frac{TP_c}{TP_c + FP_c}$ | $\frac{2TP_c}{2TP_c + FP_c + FN_c}$ | $\frac{TP_c}{TP_c + FP_c + FN_c}$ | $\frac{TP_c}{TP_c + \frac{1}{2}FP_c + \frac{1}{2}FN_c}$ | $\frac{TP_c}{TP_c + \alpha FP_c + \beta FN_c}$ |

Table 1: **Region Metrics:** We show both set representation and Boolean representation. TP, FN and FP stand for true positive, false negative and false positive respectively. The subscript $c$ means that the metrics are calculated for each class.

loss of each sample by $1-p$ where $p$ is the predicted probability/confidence. Intuitively, hard samples will have low confidence and therefore a high weight. It can be thought of as a hard-example mining loss. To see recall loss's resemblance to focal loss, we need to need to rewrite it slightly.

$$FocalCE = -\sum_{n=1}^{N}(1 - p_{n,t}^{y_n})^\gamma \log p_{n,t}^{y_n} = -\sum_{c=1}^{C}\sum_{n:y_n=c}(1 - p_{n,t})^\gamma \log p_{n,t} \qquad (6)$$

where $p_{n,t}^{y_n}$ is predicted probability of class $y_n$ for sample $n$ at time $t$; $\gamma$ is a scalar hyperparameter.

Focal loss has been a very popular loss function for imbalance in detection. It is appealing because it *dynamically* adjusts the weight for each sample depending on the difficulty of the sample and model performance. However, the focal loss is not specifically effective against imbalanced classification problems as reflected by poor performance reported by many papers (Cao et al., 2019; Cui et al., 2019). By examining the similarity between Eq. 5 and Eq. 6, we argue that the proposed recall loss function can be seen as a class-wise focal loss with $\gamma = 1$ and the per-class metric $\mathcal{R}_{c,t}$ replacing per-sample probability $p_{n,t}^{y_n}$. The next section will discuss how to estimate the recall for each class.

## 3.3 RECALL ESTIMATION

The recall loss is designed to reflect the instantaneous training performance of a model on the current input data. A straightforward way is to estimate the recall based on the current batch statistics, i.e., count false positives for each class from an entire batch. This method provides a reliable estimation of the model's current performance *if there is a sufficient number of samples for each class in the batch.* Intuitively for classification, batch recall is a good estimation if the number of classes is not much larger than the batch size. For semantic segmentation, batch recall is almost always reliable since each image contains hundreds of pixels for each class. For subsequent segmentation experiments, we use the *batch recall loss* where the batch recall is calculated as follows:

$$\mathcal{R}_{c,t} = \frac{TP_{c,t}}{TP_{c,t} + FN_{c,t}} \qquad (7)$$

For classification, estimating recall is problematic for a large number of classes. For example, the iNaturalist2018 dataset has 8,142 classes. For a batch size of 128, it is difficult to sample sufficient data for any class. To mitigate the problem, we use the Exponential Moving Average (EMA) to estimate the recall and calculate the *EMA recall loss*.

$$\tilde{\mathcal{R}}_{c,t} = \alpha \mathcal{R}_{c,t} + (1 - \alpha)\mathcal{R}_{c,t-1} \qquad (8)$$

## 3.4 ANALYSIS: RECALL, PRECISION, DICE, JACCARD AND TVESKY

Why do we not use other metrics such as F1, Dice, Jaccard and Tvesky as the weights? Following previous convention, let $\mathcal{G}_c$ and $\mathcal{P}_c$ denote the set of ground truth (positive) samples and predicted samples for class $c$. Let $FP_c, TN_c$ denote the set of false positive and true negative samples respectively for class $c$, and other terms are defined similarly. Recall is different from the other metrics in that it does not have false positive in the denominator and this distinction makes it ideal for weighting cross entropy loss (and others not) as shown in table 1. Referring back to Eq. 5, where recall loss is defined as weighted cross entropy by $1 - \mathcal{R}_c$, replacing recall by any other metrics above would result in FP appearing in the numerator of the weights. For example, a hypothetical precision loss can be defined as following.

$$PrecisionLoss - \sum_{c=1}^{C}\sum_{n:y_i=c}\left(\frac{FP_{c,t}}{FP_{c,t} + TP_{c,t}}\right)^\gamma \log(p_{n,t}) \qquad (9)$$

This formulation will introduce unexpected behavior. A large false positive count in a class will result in a large weight, which further encourages false detection for that class. This will cause the number of false positives to explode. From a different perspective, because in cross entropy loss we always penalize the ground truth samples $i \in \mathcal{G}_c = \{i : y_i = c\}$ for a class $c$, a proper weighting should be proportional to $FN_c \subseteq \mathcal{G}_c$ but not $FP_c \not\subset \mathcal{G}_c$ which does not belong to the set of ground truth samples. The same analysis can be applied to other metrics involving false positives.

### 3.5 Recall Loss as a Feature Learning Loss for Imbalanced Classification

Recent works (Kang et al., 2020) on the imbalanced classification problem proposed to separate representation learning and classifier learning. Intuitively, the final classifier layer is negatively affected by highly imbalanced data distributions while the convolutional neural network backbone is not as affected. In other words, representation learning benefits from all the data regardless of their class membership. It has been shown in Kang et al. (2020) that weighted losses do not produce large improvement because they can negatively affect representation learning. While we need to be careful when introducing weighted losses to train the feature extractor, some previous works (Cao et al., 2019; Zhou et al., 2020) showed that it can still be beneficial to carefully "fine-tune" CNNs with balancing techniques towards the end of a training cycle when the learning rate has been annealed. We propose to use recall loss as a feature learning loss to replace the standard cross entropy. We experimentally show that recall loss is a better suited loss for representation learning because it considers imbalance in datasets while dynamically adjusting the weights to not bias towards any class. To apply recall loss to classification, we introduce a Simple Decoupled Network (SDN) to decouple representation and classifier learning (fig. 1(b),(c)).

Let $f_\theta$ denote the feature extractor and $\{g_\theta, h_\theta\}$ denote two classifier heads on top of the feature extractor. Generally speaking, $f_\theta$ is parameterized by a CNN and $\{g_\theta, h_\theta\}$ are two separate fully connected networks. Similar to previous works (Kang et al., 2020), (Zhou et al., 2020), we design a simple decoupled network with two classification heads and a shared feature backbone as shown in fig. 1 (b). More specifically, the loss from the head $g_\theta$ backpropagates to the feature extractor $f_\theta$ while the loss from the head $h_\theta$ is stopped. The $g_\theta$ head is trained with recall loss and the $h_\theta$ head is trained with the CB loss (Cui et al., 2019). In other words, recall loss trains the feature extractor while the CB loss does not. We only use the $h_\theta$ head in inference. Therefore, this simple modification during training does not introduce any additional change to inference. The proposed method simplifies BBN (Zhou et al., 2020) in two ways. 1) Only one loss function affects the backbone. Therefore, there is no need for hand-tuning an adaptor function for controlling two losses. 2) We only use one head for inference and discard the other. This simple design proves to be effective in our experiments.

## 4 Experiments

### 4.1 Experimental Setting

**Datasets.** We evaluate our recall loss on two popular large-scale outdoor semantic segmentation datasets, Synthia (Ros et al., 2016) and Cityscapes (Cordts et al., 2016). Synthia is a photorealistic synthetic dataset with different seasons, weather, and lighting conditions. Specifically, we use the Synthia-sequence Summer split for training (4400), validation (624), and testing (1272). Cityscapes consists of real photos of urban street scenes in several cities in Europe. It has 5000 annotated images for training and another 5000 for evaluation. We further show that recall loss is beneficial for feature learning in image classification on two large-scale long-tailed datasets including Place-LT (Liu et al., 2019) and iNaturvspacealist2018 (Van Horn et al., 2018). Place-LT has 365 classes and long-tailed class distribution. It is created by sampling from the original balanced dataset (Zhou et al., 2017) following a Pareto distribution. iNaturalist2018 is a long-tailed image collection of natural species of 8142 classes. Please refer to Appendix A.2 for details on implementation.

**Evaluation Metrics.** We report *mean accuracy* and *mean IOU* for semantic segmentation experiments and *overall accuracy* for image classification following previous works (Cao et al., 2019; Zhou et al., 2020; Liu et al., 2019; Kang et al., 2020) on these datasts. We note that both mean accuracy and mean IOU are important metrics for semantic segmentation. While a good mean IOU

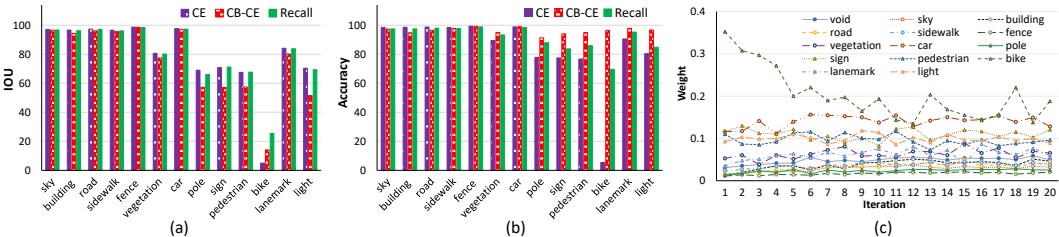

Figure 2: **(a)** Mean IOU per class on Synthia dataset with Resnet18. **(b)** Mean Accuracy per class on Synthia dataset with Resenet18. **(c)** Normalized $1 - \mathcal{R}_{c,t}$ weights over time for recall loss. The weight for the bike class decreased over time indicating improved performance.

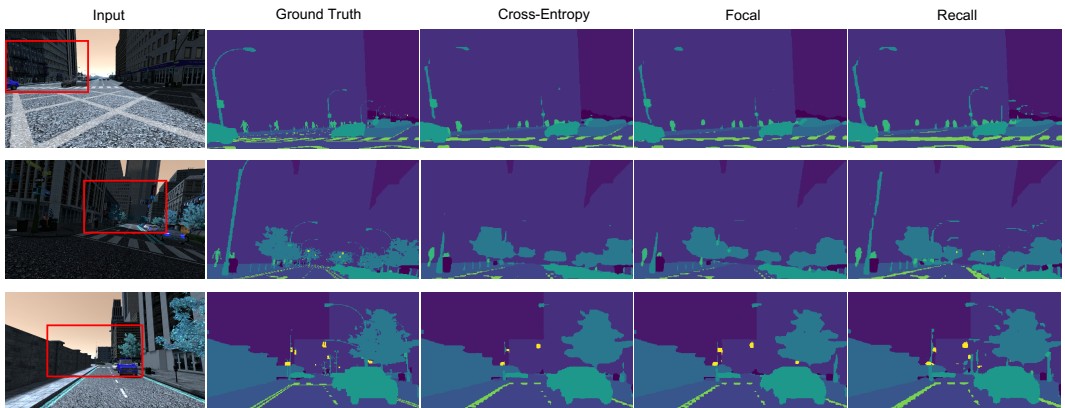

Figure 3: Visualization of Segmentation results on Synthia with Resenet18. Recall loss encourages models to predict more small classes such as poles and pedestrians. Compared to the cross-entropy trained model, the recall loss trained model is able to output finer details especially for small classes.

indicates a balanced performance of precision and recall, mean accuracy is an indicator of the detection rate of each class, which is important for safety-critical applications such as self-driving.

## 4.2 SEMANTIC SEGMENTATION EXPERIMENTS

For semantic segmentation experiments, we compare our method with two region-based losses, SoftIOU (Rahman & Wang, 2016) and Lovasz softamax (Berman et al., 2018). Both of these losses aim to minimize a soft/surrogate version of the metric IOU. As analyzed by Eelbode et al. (2020), the softDice loss (Sudre et al., 2017) and softTversky loss (Salehi et al., 2017) are similar to the two chosen losses. It is representative to compare to two of them. We also compare with state-of-the-art losses for imbalanced image classification and detection such as CB-CE (Cui et al., 2019), Focal loss (Lin et al., 2017) and Online Hard Example Mining (OHEM) loss (Shrivastava et al., 2016). We keep top $70\%$ samples in OHEM. While there are other techniques for imbalanced image classification, they require changes to learning schedules and architectures. A direct adaptation of them for semantic segmentation is not trivial. We compare with them directly on image classification datasets.

|  | CE | SoftIOU | Lovasz | OHEM | CB-CE | Focal | Recall |
|---|---|---|---|---|---|---|---|
| mIOU | 79.87 | 70.29 | 68.20 | 64.56 | 75.41 | 79.91 | 80.78 |
| mACC | 84.32 | 88.03 | 87.81 | 76.87 | 95.76 | 84.78 | 91.03 |

(a) **DeepLab Resnet18 performance on Synthia**

|  | CE | SoftIOU | Lovasz | OHEM | CB-CE | Focal | Recall |
|---|---|---|---|---|---|---|---|
| mIOU | 80.84 | 72.02 | 70.79 | 65.13 | 75.45 | 80.41 | 79.84 |
| mACC | 85.34 | 88.64 | 91.01 | 81.37 | 95.78 | 85.04 | 91.14 |

(b) **DeepLab Resnet101 performance on Synthia**

Table 2: We compare the recall loss with other losses on the Synthia dataset using two different backbone networks. Recall loss improves accuracy significantly while offers competitive IOU.

| | CE | SoftIOU | Lovasz | OHEM | CB-CE | Focal | Recall |
|---|---|---|---|---|---|---|---|
| mIOU | 60.66 | 60.49 | 55.70 | 50.00 | 36.37 | 62.11 | 60.99 |
| mACC | 72.24 | 73.51 | 67.68 | 61.41 | 68.99 | 74.47 | 77.28 |

(a) **DeepLab Resnet18 performance on Cityscapes**

| | CE | SoftIOU | Lovasz | OHEM | CB-CE | Focal | Recall |
|---|---|---|---|---|---|---|---|
| mIOU | 70.42 | 65.49 | 49.83 | 56.69 | 58.59 | 67.11 | 70.51 |
| mACC | 80.54 | 80.65 | 61.77 | 68.54 | 86.42 | 78.92 | 81.34 |

(b) **DeepLab Resnet101 performance on Cityscapes**

Table 3: We compare recall loss with other losses on the Cityscapes dataset with real images using different backbones. Recall loss yields the best accuracy and maintains a competitive IOU. This demonstrates its robustness to label noise in real datasets.

| $\alpha$ | 0.1 | 0.3 | 0.5 | 0.7 | 0.9 |
|---|---|---|---|---|---|
| Resnet152 | 40.01 | 39.43 | 40.01 | 39.56 | 39.81 |

(a) **Accuracy with different $\alpha$ on Place-LT**

| $\alpha$ | 0.001 | 0.01 | 0.1 | 0.2 | 0.3 |
|---|---|---|---|---|---|
| Resnet50 | 67.03 | 66.70 | 66.57 | 66.20 | 66.01 |

(b) **Accuracy with different $\alpha$ on iNaturalist2018.**

Table 4: To study sensitivity of EMA on different datasets, we experimented with different $\alpha$ values and report the validation accuracy. We use $\alpha = 0.9$ and $\alpha = 0.001$ for Place365 and iNaturalist2018 respectively in subsequent comparison with other methods.

**Synthia.** We first present results on the synthetic Synthia segmentation dataset in table 2. Soft-IOU, Lovasz, CB-CE all improved the mean accuracy compared to the baseline cross entropy loss. However, the improvement comes at a cost of lower mean IOU. Focal loss performs similarly to the standard cross entropy loss. This is consistent with our statement that hard-example weighting in focal loss is not effective against multi-class imbalance. OHEM performs worse on both metrics. We think this is because OHEM completely discards 30% training samples at each iteration and this negatively affects feature learning. On the other hand, the proposed recall loss improves the mean accuracy metric and maintains good mean IOU. This validates our analysis that the recall loss can balance between precision and recall. The same trend is observed for both shallow backbone, resnet18, and deep backbone resnet101. However, we note that the effectiveness of the recall loss is more obvious on a less powerful feature backbone. We hypothesize that less powerful backbones are more likely to spend its limited representation capability on large class and thus bias towards them. This is important because in many applications, hardware limits the deployment of computation-heavy backbones and we need to attend to small classes for safety relying on shallow feature extractors. In fig. 2, we show per-class accuracy and IOU performance on three losses including the recall loss. We observe that both CB-CE and recall loss can improve accuracy on small classes significantly. However, CB-CE deteriorates IOU for those classes significantly while recall loss maintains competitive performance because CB-CE uses a fixed weighting which *always* emphasizes small classes. This observation supports our claim that recall loss balances recall and precision because of its dynamic adaptability to performance. In the fig. 2(c), we show the $1 - \mathcal{R}_{c,t}$ weights on resenet18 Synthia experiment. We observe that the weight for small classes such as bike decreases over time. This indicates that the performance of the bike class has increased. We further provide a visual comparison between a model trained with the cross-entropy loss and the proposed recall loss in fig. 3. Our method provides more fine details on small classes which are often suppressed in traditional cross entropy training.

**Cityscapes.** We further present results on the Cityscapes segmentation dataset with real images. As shown in table 3, softIOU, and focal loss perform similarly to the standard cross entropy loss while Lovasz, OHEM, and CB-CE are consistently worse on both of the resnet backbones. Compared to performance on Synthia, this shows that some of the losses are not robust to label noise in a real dataset. Recall loss again outperforms other losses by improving mean accuracy and maintaining a good mean IOU. In other words, recall loss improves the detection rate of small classes such as pedestrians, light poles on the road and maintains a good precision. This demonstrates the effectiveness of recall loss on both synthetic and real datasets on outdoor driving segmentation datasets.

### 4.3 IMAGE CLASSIFICATION EXPERIMENTS

For classification experiments, we compare on two popular imbalanced datasets, Place-LT (Liu et al., 2019) and iNaturalist2018 (Van Horn et al., 2018). We compare to the methods that have achieved state-of-the-art performance on either dataset. We specifically introduce a baseline model, SDN-CE, which replaces recall loss with a standard cross entropy loss to train the feature backbone.

| | CE | Lifted Loss† | Focal Loss † | Range Loss † | OLTR † | tau-norm ‡ | LWS‡ | SDN (CE) | SDN (recall) |
|---|---|---|---|---|---|---|---|---|---|
| All | 30.6 | 35.20 | 34.60 | 35.10 | 35.9 | 37.9 | 37.6 | 39.3 | **39.8** |
| Many | **46.7** | 41.1 | 41.1 | 41.1 | 44.7 | 37.8 | 40.6 | 43.0 | 43.3 |
| Medium | 30.1 | 35.4 | 34.8 | 35.4 | 37.0 | 40.7 | 39.1 | 40.3 | **41.0** |
| Few | 11.9 | 24.0 | 22.4 | 23.2 | 25.3 | **31.8** | 28.6 | 31.1 | 31.2 |

Table 5: **Test Accuracy on Place-LT.** † denotes results from Liu et al. (2019) and ‡ denotes results from Kang et al. (2020). We follow previous works to report performance on many (more than 100 images), medium (20 to 100 images) and few-shot (fewer than 20 images). Our method yields overall best performance. We highlight the **Best** and Second Best baseline method.

| | CE | Focal | Focal-DRW | LDAM | LDAM-DRW† | BBN† | tau-norm† | LWS† | SDN (CE) | DN (Recall) |
|---|---|---|---|---|---|---|---|---|---|---|
| Resnet50 | 62.00 | 61.67 | 65.62 | 60.38 | 64.58 | 66.29 | 65.60 | 65.90 | 66.13 | **67.03** |

Table 6: **Validation Accuracy on iNaturalist2018.** † denotes results from Zhou et al. (2020) and ‡ denotes results from Kang et al. (2020).

**Place-LT.** Following previous works (Kang et al., 2020), (Liu et al., 2019), we compare to the Lifted loss (Oh Song et al., 2016), Focal loss (Lin et al., 2017), Range Loss (Zhang et al., 2017), OLTR (Liu et al., 2019), tau-norm (Kang et al., 2020) and LWS (Kang et al., 2020). We note that Kang et al. (2020) experimented with many variants and tau-norm is the best performing one on this dataset. Table 5 shows that the recall loss outperforms other loss including SDN-CE. The result is two-fold. First, the strong performance of the baseline, SDN-CE agrees with the finding in Kang et al. (2020) that imbalance affects the classifier more than the backbone and a simple decoupling trick can outperform carefully designed losses. Second, the result validates our claim that the recall loss, SDN-recall, is a more suitable feature loss for imbalanced datasets when comparing to SDN-CE. Note that we use the EMA version of the recall loss. The table 4(a) shows the results of SDN-recall with different $\alpha$ on a validation set. We can conclude that the sensitivity of $\alpha$ is low on the Place-LT dataset. Specifically, the ratio of the number of classes to batch size is 365:128 in this case.

**iNaturalist2018.** On iNaturalist we compare to LDAM-DRW (Cao et al., 2019), BBN (Zhou et al., 2020), tau-norm (Kang et al., 2020) and LWS (Kang et al., 2020). This dataset is the most challenging due to its extremely large number of classes. This presents a specific challenge to recall loss since the effectiveness of recall loss depends on a reliable estimation of the training recall for each class. Consequently, it motivated us to propose the exponential-moving average update rules. Table 6 shows the results of SDN-recall, SDN-CE, and all compared methods. SDN-recall outperforms all other methods including SDN-CE. It is worth mentioning that both LDAM-DRW and BBN proposed to finetune the feature extractor with a balanced loss using a two-stage and two-branch strategy respectively. Recall loss trains a feature backbone in an end-to-end manner and outperforms other techniques that require careful hyperparameter tuning or modification to the architectures. Table 4(b) shows the sensitivity of $\alpha$ on this dataset. As the number of classes (8,142) is much larger than the batch size (128), a small $\alpha$ is critical to provide a reliable recall estimation. We also present the training curves with different $\alpha$ in the Appendix A.3. We observe that smaller $\alpha$ yields lower training loss.

## 5 CONCLUSION

In this paper, we proposed a novel loss function based on the metric recall. The loss function uses a *hard-class mining* strategy to improve model performance on imbalanced datasets. Specifically, the recall loss weighs examples in a class based on its instantaneous recall performance during training, and the weights change dynamically to reflect *relative* change in performance among classes. Experimentally, we demonstrated several advantages of the loss: 1) Recall loss improves accuracy while maintains a competitive IOU performance in semantic segmentation. Most notably, recall loss improves both accuracy and precision significantly in small networks, which possesses limited representation power and is more prone to biased performance due to data imbalance. 2) Recall loss works on both synthetic and real data and is robust to label noise present in real datasets. 3) The EMA version of recall loss is able to handle an extremely large numbers of classes and provides a stable improvement on representation learning. 4) Recall loss facilitates representation learning in image classification. Using a simple decoupled training strategy and the recall loss, we outperform more complicated methods on common imbalance learning benchmarks.

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
