# OpenReview forum: "Recall Loss for Imbalanced Image Classification and Semantic Segmentation"
_ICLR.cc/2021/Conference — Reject_

### Official Review · AnonReviewer3 · 2020-10-28
**In this manuscript, the authors proposed a recall loss to solve the class-imbalance issue in the image classification and segmentation tasks. The recall loss assigns weights for each class in the traditional cross-entropy loss based on the training performance under the recall metric.**

**Rating:** 5
**Confidence:** 3

**Review:**

Strength:

1- The problem of class imbalance issue studied in this work is important and necessary for general image analysis tasks.

2- On the image classification tasks for Place-LT and iNaturalist2018, the proposed method achieves state-of-the-art performance. This further demonstrates the effectiveness of the proposed method.

3- On the image segmentation tasks, the proposed method achieves the best mean accuracies and competitive mean IoU in comparison with several typical loss functions.

4- The overall paper is clearly presented and easy to follow. The motivation introduced in Section 3 provides the readers with a detailed illustration of the importance of the method. The clearly described motivations induce the readers to better and more clearly understand the overall work.


Weakness:

1- The first concern of the proposed method is unstable performance. For the image segmentation tasks, the overall performance of the proposed method is not stable and robust. For example, the method sometimes achieves worse mIoU in comparison with Focal loss (as shown in Table 1-b and Table 2-a). Although the proposed method always achieves the best mACC metric, the less competitive performance under mIoU score still limits the overall performance of the proposed method.

2- The second concern is the lack of comparison with the state-of-the-art methods on the semantic segmentation tasks. Although the proposed method is demonstrated to be effective compare with recent loss functions, it is not convinced without a competitive performance compared with recent state-of-the-art semantic segmentation methods.


3- There is a lack of visualization comparison results among each method. As a paper based on image analysis, the visualization comparison experiment is also important and necessary. This could be easily addressed but should be included nonetheless.

---

> ### Author Response · Authors · 2020-11-13
> **We addressed concerns and added visualization suggested by the reviewer.**
>
> We thank the reviewer for constructive feedback on experiments and discussion on performance. We also appreciate the reviewer for highlighting the importance of the imbalance problem and positive comments on the content of the paper.  In this paper, we proposed a loss function to improve image classification and segmentation. For image classification, we aim to improve overall accuracy and for segmentation, we aim to improve mean accuracy while maintaining a competitive IoU.
>
> **1) Regarding performance**\
> IoU and accuracy are a trade-off much like precision and recall. Normally, improving on one degrades the other.  The proposed loss can significantly improve accuracy performance *consistently* (mostly from small classes) while maintaining a good IOU performance on Synthia and Cityscape. For example, in table 2 (a), compared to cross-entropy loss, the recall loss improved mean accuracy by 8.3% relatively while maintaining a competitive IOU performance. If we only count the small classes (pole, sign, pedestrian, bike, light), recall loss improved 30% relatively on mean accuracy and improved 6% on mean IOU relatively. The drop on large classes is marginal (-0.048% in mean accuracy and -0.038% in mean IOU) compared to the improvement we could obtain from the small classes. The other compared losses including cross-entropy degrade accuracy in favor of IoU.  We believe this improvement should not be ignored for safety considerations. For example, we might care more about pedestrians and poles on the road than a slight drop in IOU in the background. As stated in this paper [1], “the relative importance assigned to precision and recall should be an aspect of the problem”. We believe the significant improvement in accuracy (recall) outweighs no/marginal drop on mIoU. Further, we newly added visualization for segmentation in the paper as part of this discussion. Figure 3 in the revised main paper for the rebuttal shows that our method can produce finer details for small classes. We would also like to point out that our method *consistently* achieves state-of-the-art accuracy on large-scale imbalance classification datasets as well (iNaturalist2018, Place365). '
>
> [1] Hand D, Christen P. A note on using the F-measure for evaluating record linkage algorithms[J]. Statistics and Computing, 2018
>
> **2) Regarding comparison**\
> For semantic segmentation, while we adopted a state-of-the-art segmentation network, DeepLabV3, we focused on relative performance improvement using different loss functions. This is essentially a controlled experiment in which the only variable is the loss function for fair comparison and comparability. Competing absolutely with state-of-the-art performance requires extensive parameter tuning for all compared methods and our proposed methods. We fixed the same set of training parameters for all methods and conducted a controlled experiment to identify the strength of our loss on semantic segmentation. We aimed to demonstrate performance differences with minimum tuning. State-of-the-art segmentation models also suffer from class imbalance [1] resulting in ignoring important small classes and class-balancing is not widely applied to them [1]. One reason is that class-imbalance does not affect IoU significantly which is the main metrix people use. However, this can cause models to ignore small classes (accuracy) and result in potential safety issues especially for self-driving.  The proposed loss function can improve the accuracy of the small networks significantly and incur no/marginal drop on mIOU. For image classification, the loss function together with the SDN training strategy showed state-of-the-art performance on two large-scale imbalance classification datasets compared to 11 competing methods.
>
> [1] Chan R, Rottmann M, Hüger F, et al. Application of decision rules for handling class imbalance in semantic segmentation[J]. arXiv preprint arXiv:1901.08394, 2019.
>
> **3) Regarding visualization**\
> Thank you for the great suggestions. We now provide visualization for the segmentation task in figure 3 in the revised main paper.

---

### Official Review · AnonReviewer1 · 2020-10-28
**Review-Recall loss for imbalanced problems**

**Rating:** 6
**Confidence:** 5

**Review:**

This paper proposes a new loss based on the Recall metric to deal with imbalance problems in several visual recognition tasks (i.e., classification and segmentation). Authors show, in several public benchmarks, that the proposed recall loss outperforms other losses in the tasks of classification and segmentation. Please find below my comments.

Strengths.

-The paper is easy to follow.
-Proposing new losses to deal with imbalance problems is an interesting venue.
-Results show that the proposed loss outperforms the other losses compared in the paper.

Weaknesses.

-The methodological contribution is marginal/incremental. The proposed loss is a straightforward extension of prior losses.

-More fundamentally, I wonder about the motivation of this loss wrt to the use of other metrics as losses. For example, the F1-score of a class is given by the harmonic mean of precision and recall, combining both. Why a recall loss is better than an F1-based loss? For example, the paper in [1] (not included in related work), propose a novel balance function balancing these two terms in the context of highly unbalanced segmentation. Related to this, if model predictions obtain low recall and low precision for a given class, that class is poorly handled by the model, which again motivates the use of something beyond just the recall as objective function.

-There exist missing literature in the losses to deal with imbalance. Particularly, recent works (e.g., [2]) have introduced boundary-based losses for segmentation, which deal with the problems of region-based losses (e.g., large gradients or issues to weight classes bases on their frequency). Authors should discuss these papers, as well as include them in the results.

-I also believe the experiments are poorly conducted. Authors resort to mean-IoU and mean-Acc for segmentation, and accuracy for classification. Nevertheless, F1-score is a popular choice on imbalance problems. Authors should reconsider other metrics to better show the impact of the proposed loss. Furthermore, in both classification and segmentation, the results are averaged over all the classes, when comparing to prior work (not just CE and CBCE). This makes that one does not know which is the impact of the proposed loss in the under-represented classes. Is the improvement due to the model improves the performance over all the classes? Or is it actually coming because the model better handles minority classes? With the current results, it is not possible to evaluate the contribution of this loss in imbalance problems.

-Related to my previous comment, authors only show the per-class mIOU and Acc on the segmentation task, and comparing to CE and CB-CE (Fig. 2a and 2b). Looking at the Fig 2a, which is the metric typically employed to evaluate the segmentation task on these datasets, we can observe that the proposed loss only improves one class over the standard CE (i.e., bike), falling behind the standard CE in all the rest. These results, together with the values reported in Tables 1 and 2 (where CE outperforms the proposed loss in 1 case, and in the other 3 cases the improvement over CE is marginal), make me wonder about the usability of the proposed loss.

[1] Hashemi et al. Assymmetric similarity loss function to balance precision and recall in highly unbalanced deep medical image segmentation IEEE 2019.
[2] Kervadec et al. Boundary loss for highly unbalanced segmentation. MIDL 2019.

---

> ### Author Response · Authors · 2020-11-13
> **We have reviewed the comments and suggested papers.**
>
> We thank the reviewer for constructive feedback on the metrics and writing of the paper. We have revised the papers in terms of formatting after reading the comments.
>
> **1) Regarding contribution**\
> While we draw an analogy between the proposed recall loss and focal loss/inverse frequency loss analytically to help the reader clarify the relationships between the proposed loss and other widely-used functions, we do not consider the loss function as an extension of prior loss functions. It is not built upon the formulation of any other losses. We motivated the weighting scheme based on the performance recall metrics and this is different from all previous losses. However, we would like to point out that the idea of weighting cross-entropy is not new. It has been explored by many papers [1][2][3]. They innovated the weighting schemes for cross-entropy loss. While we do not claim novelty to weighting cross-entropy, using instantaneous training recall (batch/EMA) is a novel idea. We further explored advantages including dynamism and balanced-performance analytically and empirically.
>
> [1] Lin T Y, Goyal P, Girshick R, et al. Focal loss for dense object detection ICCV ( 2017)
>
> [2] Cui Y, Jia M, Lin T Y, et al. Class-balanced loss based on effective number of samples CVPR (2019)
>
> [3] Cao K, Wei C, Gaidon A, et al. Learning imbalanced datasets with label-distribution-aware margin loss NeurIPS (2019)
>
> **2) Regarding motivation**\
> This is a great question that we actually dedicated a whole section in the paper Sec.3.4 (originally Section A.2 in the supplementary/appendix document). We empirically found that F1-based losses were not ideal for weighting cross entropy loss. To obtain more insights, we investigated the problem analytically in Sec.3.4. There is a range of similar metrics linked to F1 such as Dice, Jaccard, Tversky. Please see table 1 in the main paper for the comparison. Notably, the Tversky score is the general form of the F1 score and F1 score is equivalent to the Dice score. The mathematical reason why F1 score is not suitable is that it has a False Positive (FP) factor in the denominator. Weighting by F1 score will result in the opposite effects i.e., a large class with more false positive detection will be further encouraged. This is exactly the opposite weighting we would like to impose on the loss function. The same analysis applies to all the mentioned metrics including F1 and leaves recall as the only metric suitable to tackle imbalance. Again, please refer to Sec.3.4  in the main paper for more detailed analysis. There are many papers in the medical image segmentation field using region-based losses. As we did an extensive search in that field, all of the metrics (Dice[1], Jaccard[2][3], Tversky[4]) have been widely used as metrics and loss functions.  As pointed out by [5], they do not show very different performance and we also compared with two of them ([2][3]) which directly optimize the metric IOU. They tend to encourage accuracy but not precision (table 1 and 2). It is worth pointing out that these are region-based losses developed specifically for semantic segmentation not for image classification while our loss function can be used on both.
>
> [1] Sudre C H, Li W, Vercauteren T, et al. Generalised dice overlap as a deep learning loss function for highly unbalanced segmentations Deep learning in medical image analysis and multimodal learning for clinical decision support. (2017)
>
> [2] Berman M, Rannen Triki A, Blaschko M B. The lovász-softmax loss: A tractable surrogate for the optimization of the intersection-over-union measure in neural networks CVPR (2018)
>
> [3] Rahman M A, Wang Y. Optimizing intersection-over-union in deep neural networks for image segmentation International symposium on visual computing. Springer (2016)
>
> [4] Salehi S S M, Erdogmus D, Gholipour A. Tversky loss function for image segmentation using 3D fully convolutional deep networks International Workshop on Machine Learning in Medical Imaging (2017)
>
> [5] Eelbode T, Bertels J, Berman M, et al. Optimization for medical image segmentation: Theory and practice when evaluating with dice score or jaccard index  IEEE Transactions on Medical Imaging (2020).

---

> > ### Author Response · Authors · 2020-11-13
> > **Cont’d**
> >
> > **3) Regarding missing literature**\
> > Thank you for bringing this up. We are happy to cite and discuss these interesting works.  As in many medical imaging works, the boundary loss [1] is limited to binary (two-region) applications such as lesion segmentation. As stated in the boundary loss paper, an extension to multi-region is their future work. The segmentation dataset we tested on is a multi-class multi-region outdoor self-driving dataset. Therefore, this work cannot be directly compared to either semantic segmentation or image classification. However, we did compare to two popular loss functions ([1],[2]) from the medical imaging field. These two losses directly optimize IOU which is the most popular metric for segmentation.
> >
> > [1] Hashemi S R, Salehi S S M, Erdogmus D, et al. Asymmetric similarity loss function to balance precision and recall in highly unbalanced deep medical image segmentation (2018).
> >
> > [1] Berman M, Rannen Triki A, Blaschko M B. The lovász-softmax loss: A tractable surrogate for the optimization of the intersection-over-union measure in neural networks CVPR (2018)
> >
> > [2] Rahman M A, Wang Y. Optimizing intersection-over-union in deep neural networks for image segmentation International symposium on visual computing. Springer (2016)
> >
> > **4) Regarding experiments and metrics**\
> > Following previous works on semantic segmentation[1][2][3] and imbalanced classification [4][5][6][7], we decided to use mean-IOU and mean-accuracy for segmentation and accuracy for classification.
> >
> > 1) Semantic Segmentation: In segmentation, both training and testing data are imbalanced because of uneven pixel percentage of different classes. Therefore, it makes sense to use a metric that considers both precision and recall. **While we considered using F1 score, it is very closely related to the metric IOU (Jaccard).** They convey essentially the same message in terms of precision and recall. While this could be different customs in different fields, the more common metric in the computer vision community is IOU ([1][2][3]). Please see table 1 in the main paper for the comparison between these metrics.
> >
> > 2) Image Classification: We want to clarify that **the test sets are all uniformly distributed and imbalance only exists in the training sets.**  F1 score is preferred when the data is highly imbalanced, which is not the case in our test sets, Place365-LT, and iNaturalist2018 we used. Also, in our case, accuracy is equivalent to mean accuracy. Moreover, the use of F1 score to judge the performance of a classifier is debatable [8,8] because it assigns the same weight to recall and precision. This could be misleading since “the relative importance assigned to precision and recall should be an aspect of the problem” [8]. Whether using F1 or not is a specific aspect of a problem and is not necessarily better than accuracy for some problems.
> >
> > We thank the reviewer for the thoughtful comments on experiments. To obtain insights into the performance gain of the method, we added more quantitative and qualitative results. It is clear that the proposed loss improves performance on small classes significantly while not sacrificing performance on large classes in both segmentation and classification.
> >
> > 1) For segmentation, we showed the per-class performance in figure 5 for segmentation tasks. We observed that recall loss improves accuracy on the small classes most notably. Please refer to Sec.4.2 Synthia experiment for more detailed description. In the revised version as part of this rebuttal, we also added visualization in figure 3 in the main paper. The visualization shows that our method can produce fine details for small classes which are ignored by other methods.
> >
> >
> > [1] Long J, Shelhamer E, Darrell T. Fully convolutional networks for semantic segmentation CVPR (2015)
> >
> > [2] Badrinarayanan V, Kendall A, Cipolla R. Segnet: A deep convolutional encoder-decoder architecture for image segmentation[J]. IEEE transactions on pattern analysis and machine intelligence ( 2017)
> >
> > [3] Chen L C, Papandreou G, Kokkinos I, et al. Deeplab: Semantic image segmentation with deep convolutional nets, atrous convolution, and fully connected crfs[J]. IEEE transactions on pattern analysis and machine intelligence, (2017)
> >
> > [4] Cui Y, Jia M, Lin T Y, et al. Class-balanced loss based on effective number of samples CVPR (2019):
> >
> > [5] Cao K, Wei C, Gaidon A, et al. Learning imbalanced datasets with label-distribution-aware margin loss NeurIPS (2019)
> >
> > [6] Kang B, Xie S, Rohrbach M, et al. Decoupling representation and classifier for long-tailed recognition ICLR (2019)
> >
> > [7] Zhou B, Cui Q, Wei X S, et al. BBN: Bilateral-Branch Network with Cumulative Learning for Long-Tailed Visual Recognition CVPR (2020)
> >
> > [8] Hand D, Christen P. A note on using the F-measure for evaluating record linkage algorithms[J]. Statistics and Computing (2018)
> >
> > [9] Powers D M W. What the F-measure doesn't measure: Features, Flaws, Fallacies and Fixes[J]. (2015)

---

> > > ### Author Response · Authors · 2020-11-13
> > > **(Cont’d)**
> > >
> > > 2) For classification, we expanded Table 5 (Place365 dataset) in the main paper to breakdown performance into 3 categories: *many* (more than 100 images), *medium* (20 to 100 images), and *few* (fewer than 20 images) based on the training set. This should provide further intuition into where the performance gain is from. Our method yields the overall best performance by improving performance on *medium* and *few* while maintaining a good *many* performance.
> > >
> > > We reproduce table 5 here for easy access.
> > >
> > > ----------------------------------------------------------------------------------------------------------------------------------------\
> > > &emsp;&emsp;&emsp;&emsp;                 CE  &nbsp; Lifted Loss  &nbsp; Focal Loss &nbsp;  Range Loss  &nbsp; OTLR  &nbsp; Tau-norm &nbsp;   LWS &nbsp; | SDN(CE)  &nbsp;SDN (Recall)\
> > > ----------------------------------------------------------------------------------------------------------------------------------------\
> > > All           &ensp;&emsp;&emsp;30.6     &emsp;&emsp;35.2              &emsp;&emsp;&emsp;34.6              &emsp;&emsp;&emsp;&emsp;35.1         &nbsp;&emsp;&emsp;35.9         &emsp;&emsp;37.9      &emsp;&emsp;37.6   |     &emsp;&emsp;39.3              &ensp;&emsp;&emsp;**39.8**\
> > > Many      &emsp;**46.7**      &emsp;&emsp;41.1              &emsp;&emsp;&emsp;41.1              &emsp;&emsp;&emsp;&emsp;41.1          &nbsp;&emsp;&emsp;44.7         &emsp;&emsp;37.8       &emsp;&emsp;40.6  |      &emsp;&emsp;43.0             &ensp;&emsp;&emsp;43.3\
> > > Medium   &nbsp;30.1      &emsp;&emsp;35.4             &emsp;&emsp;&emsp;34.8              &emsp;&emsp;&emsp;&emsp;35.4          &nbsp;&emsp;&emsp;37.0         &emsp;&emsp;40.7       &emsp;&emsp;39.1   |     &emsp;&emsp;40.3              &ensp;&emsp;&emsp;**41.0**\
> > > Few         &nbsp;&ensp;&emsp;11.9      &emsp;&emsp;24.0              &emsp;&emsp;&emsp;23.2             &emsp;&emsp;&emsp;&emsp;25.3          &nbsp;&emsp;&emsp;**31.8**        &emsp;&emsp;28.6       &emsp;&emsp;37.6   |      &emsp;&emsp;31.1              &ensp;&emsp;&emsp;31.2\
> > > ----------------------------------------------------------------------------------------------------------------------------------------\
> > > **Table 5**: Test Accuracy on Place-LT.   We follow previous works to report performance on *many* (more than100 images), *medium* (20 to 100 images), and *few*-shot (fewer than 20 images) based on the training set (Note that the evaluation data is uniformly distributed).  Our method yields the overall best performance
> > >
> > > **5) Regarding usability**\
> > > In this paper, we exposed a problem for semantic segmentation. The detection rate for small classes such as pedestrians, bikes, are very poor. Even though cross-entropy training results in good IOU, the accuracy of small classes is very low. This is especially bad for safety consideration because small classes are sometimes more important [1]. Therefore, we want to improve accuracy performance while maintaining a competitive IOU. The proposed loss can significantly improve accuracy performance (mostly from small classes) while maintaining a good IOU performance. Figure 2 (a) demonstrated that recall loss does not sacrifice IoU while CB-CE does. Figure 2 (b) showed that recall loss improves accuracy (recall) for small classes significantly. In table 2 (a), compared to cross-entropy loss, the recall loss improved mean accuracy by 8.3% relatively while maintaining a competitive IOU performance. If we only count the small classes (pole, sign, pedestrian, bike, light), recall loss improved 30% relatively on mean accuracy and improved 6% on mean IOU relatively (fig.2 (a) (b)). The drop in large classes is marginal (-0.048% in mean accuracy and -0.038% in mean IOU) compared to the improvement we could obtain from the small classes.  We believe that the recall loss will be important for segmentation models for safety considerations as it improves the detection rate for small classes significantly while not degrading large class performance. Please see the newly added figure 3 in the main paper. We visualized the qualitative improvement we can obtain by using the recall loss. In summary, recall loss is meant to improve accuracy (recall) for all classes while maintaining a good IoU. This means that recall loss introduces more true positives than false positives. Again, the usage of recall loss is not limited to semantic segmentation; it also demonstrated state-of-the-art performance on large scale imbalance classification datasets (iNaturalist and Place365) in table 5 and 6.
> > >
> > > [1] Chan R, Rottmann M, Hüger F, et al. Application of decision rules for handling class imbalance in semantic segmentation[J]. arXiv preprint arXiv:1901.08394, 2019.

---

### Official Review · AnonReviewer2 · 2020-10-28
**Novel, intuitive, and clean idea with good results. Not so convincing in certain aspects.**

**Rating:** 6
**Confidence:** 4

**Review:**

A novel recall loss (RecallCE) that considers dynamically-changing class recalls is proposed in this paper to mitigate class imbalance in long-tailed recognition problems. The class recalls are estimated using either the current batch statistics or an exponential moving average, depending on the number of class (or class diversity) present in training batches. Relationships between RecallCE and existing widely-used loss functions are mathematically shown. RecallCE performs competitively with existing loss functions on semantic segmentation tasks and outperform them on image classification tasks.

Paper strengths
- The paper does a good job introducing the proposed loss by gradually progressing from conventional CE to InvCE, and from InvCE to RecallCE. And it also relates RecallCE to Focal loss. This provides a clear picture to readers and allows readers to understand RecallCE more intuitively. The writing is generally good.
- RecallCE requires almost no hyperparameters for tuning. For a training batch that includes all the classes (e.g., Cityscapes), RecallCE has no hyperparameter. Otherwise, RecallCE only requires $\alpha$ for EMA which has been shown to robust against a good range of values in Table 3.
- The experimental results are pretty good, except on Synthia dataset where it does not always outperform other methods.

Paper weaknesses
- The choice of using/not using SDN seems arbitrary and is not motivated by anything. In the experiments, image classification tasks use SDN while semantic segmentation does not seem to be using SDN (there is no explicit statement indicating whether semantic segmentation uses SDN). Semantic segmentation is a pixel wise classification task which is still essentially a classification task that should benefit from SDN.
- There are no indications on how the hyperparameters for other loss functions are tuned (or if they use only default hyperparameters) and how they compare with RecallCE if the hyperparameters are well-tuned.
- One closely-related prior paper is not cited/discussed or compared with. This prior paper introduces Seesaw loss which dynamically adjusts class weights based on accumulated/seen training samples. While this is different from RecallCE, there is some degree of similarity. It was made available on arXiv some time before ICLR deadline.
> - Seesaw Loss for Long-Tailed Instance Segmentation (arXiv. Aug23) https://arxiv.org/abs/2008.10032
- LVIS instance segmentation dataset commonly used for evaluating loss functions designed for long-tailed recognition is not considered in this paper. Due to a large number of classes in LVIS, it is a more challenging and convincing task (since the long-tailed class distribution problem is usually more serious with a large number of classes) than Synthia and Cityscapes semantic segmentation datasets which only have relatively few classes.
- Code is not provided in the submission for reproducibility and there is no promise of code release.

Minor comments
- $L$ in Equation 4 is not consistent with the naming of other loss functions ($CE, InvCE, FocalCE, RecallCE$).
- It would be good to explain what $n:y_n=c$ stands for. Not all readers can understand this.

Overall, this paper is good but the authors need to address some of the weaknesses to make it more convincing.

=== post-rebuttal comments ==

I maintain my pre-rebuttal rating. Although the authors address my concerns reasonably well in the rebuttal, they are mostly not reflected in the paper revision (particularly on the choice of using SDN, which is quite an important piece of information for readers). There were no attempts to make a comparison with Seesaw loss, evaluating on LVIS, and releasing code anonymously (as other ICLR submissions did).

---

> ### Author Response · Authors · 2020-11-13
> **We have reviewed suggested papers by the reviewer and added citation in the revised paper.**
>
> We thank the reviewer for pointing out related work and discussion on other tasks that could benefit from our method. We appreciate the reviewer’s positive comments on the writing and that RecallCE does not introduce hyperparameters.
>
> **1) Regarding performance on Synthia**\
> While the majority of segmentation papers use Intersection over Union (IoU) as the metric. We found that it does not give a complete picture. In many applications, accuracy (recall) for small classes is more important for safety considerations [1].  High recall for pedestrians and light poles can be critical for self-driving applications.  Often, IoU and accuracy are a trade-off much like precision and recall. Improving on one degrades the other. However, recall loss can improve accuracy significantly while not degrading IoU. Figure 2 (a) demonstrated that recall loss does not sacrifice IoU while CB-CE does. Figure 2 (b) showed that recall loss improves accuracy (recall) for small classes significantly. In table 2 (a), compared to cross-entropy loss, the recall loss improved mean accuracy by 8.3% relatively while maintaining a competitive IOU performance. If we only count the small classes (pole, sign, pedestrian, bike, light), recall loss improved 30% relatively on mean accuracy and improved 6% on mean IOU relatively (fig.2 (a) (b)). The drop in large classes is marginal (-0.048% on mean accuracy and -0.038% on mean IOU) compared to the improvement we can obtain from the small classes. Overall, recall loss always provides the most balanced performance on all datasets with significantly improved accuracy and no/marginal drop on IoU. We now provide a new visualization for Synthia segmentation in figure 3 in the revised paper. The figure shows that recall loss improves details on small classes while other methods ignore them.
>
> [1] Chan R, Rottmann M, Hüger F, et al. Application of decision rules for handling class imbalance in semantic segmentation[J]. arXiv preprint arXiv:1901.08394, 2019.
>
> **2) Regarding the choice of using SDN**\
> This is a great question. To clarify, we only use SDN for image classification. The reason is rooted in the different metrics used in both tasks. In semantic segmentation, popular metrics are mean IoU and mean accuracy. Mean IoU considers both precision and recall while mean accuracy is equivalent to recall. In image classification, popular metrics are accuracy (mean accuracy). In the case of uniform test data, accuracy is equivalent to mean accuracy. In our experiments, we found that recall loss provides a good trade-off between precision and recall. It improves recall, especially in small classes, while not sacrificing precision in segmentation tasks. However, it does not always provide the best accuracy performance (Sec.4.2 Synthia). In classification where we care mostly about the accuracy, we decide to use a decoupled two branch design (SDN). The recall loss is responsible for representation learning and the Class-balanced loss is responsible for classifier learning. We showed that recall loss is better for imbalanced representation learning (SDN-CE vs. SDN-Recall). Prior works [1][2] showed that the Class-balanced loss can be detrimental to representation learning, while our method does not result in such degradation. In summary, SDN-Recall primarily focuses on improving accuracy alone while recall loss improves accuracy significantly while preserving precision in segmentation tasks.
>
> [1] Kang B, Xie S, Rohrbach M, et al. Decoupling representation and classifier for long-tailed recognition ICLR (2019)
>
> [2] Zhou B, Cui Q, Wei X S, et al. BBN: Bilateral-Branch Network with Cumulative Learning for Long-Tailed Visual Recognition CVPR (2020)
>
> **3) Regarding hyperparameters**\
> Besides the focal loss, most losses we compared in semantic segmentation do not have hyper-parameters to tune. For fair comparison, we used the same optimization hyperparameters for training all methods (learning rate, decay and epochs). We used alpha=1 and gamma=1 for focal loss in our experiments for both segmentation and classification. For image classification, we cited the optimally tuned results from prior works for fair comparison if indicated. We also reproduced some results directly using the code and hyperparameters from prior works (e.g, LDAM)

---

> > ### Author Response · Authors · 2020-11-13
> > **Cont’d**
> >
> > **4) Regarding Seesaw loss**\
> > Thank you for pointing it out; we are happy to discuss and cite this paper. Instance segmentation is an interesting area. We are very eager to test our method on object detection and instance segmentation as future work. As will be explained in the next question, we aimed to tackle two types of imbalance, inherent pixel imbalance in semantic segmentation and class imbalance in long-tailed classification.  Even though RecallLoss is different from the Seesaw loss, they have similar properties: 1) weights are dynamic, 2) both do not rely on the class distribution. However, the major difference is that the Seesaw loss still relies on cumulative training sample ratios. We discovered that a rare class is not necessarily a hard class (e.g., the fence class in Synthia). A frequency-based penalty might unnecessarily assign a high weight to an easy class which could be rare.
> >
> > **5) Regarding LVIS**\
> > In this paper, we aimed to tackle two types of imbalance. For image classification, the iNaturalist2018 has 8,412 classes with a long-tailed distribution. For segmentation, we aimed to explore the imbalance caused by the inherent difference in pixel percentage, i.e., pedestrians are much smaller in terms of the number of pixels compared to the background. The LVIS dataset is definitely an interesting benchmark because it incorporates both imbalance types we explored in this paper; we agree it would be interesting to leverage our ideas for this dataset as future work
> >
> > **6) Regarding code release**\
> > Yes, we will release the code and are strong proponents (and have a history) of open source.

---

### Official Review · AnonReviewer4 · 2020-10-30
**A good work overall, some details can be improved**

**Rating:** 7
**Confidence:** 5

**Review:**

The paper presents a learning approach to tackle class-imbalance, resulting in a good performance on frequent as well as infrequent classes. The main idea is to use the recall performance of class to adaptively weight it, such that the performance is balanced across the categories in a dataset. The paper is overall presents a strong set of ideas, is clearly presented with several performance improvements. Especially, I believe this work to be significant for segmentation literature where not many papers have shown good improvements in terms of imbalanced learning.

Pros:

+ The loss can dynamically adapt during the training process based on the recall rates.

+ Exponential moving average is used for a feasible computation of recall with a large number of classes.

+ The paper is clearly written and the loss formulations seem sound.

+ Good performance is demonstrated on classification and segmentation tasks.

+ Some of the drawbacks of statistics-balanced losses can be avoided with the use of proposed loss, e.g., reduction in the false positives compared to CB loss.

+ Good analysis is provided with other metrics and ablation studies.

+ The proposed loss is claimed to improve feature learning capability in image classification.  This is demonstrated via a Simple Decoupled Network (SDN)

Cons:

- The overall approach seems to be heavily inspired by BBN (Zhou et al., CVPR'20), especially the SDN part and the fact that a coupling adaptor function is replaced with the recall measure. This is not necessarily a negative thing in itself, however, it is not clear to me if the benefit for representation learning is due to the decoupled design or the proposed recall loss. Can the authors comment on this? Also, given the similarity with BBN, I expected a more thorough comparison with Zhou et al, they are compared against only in Table 5.

- The idea to use performance rates (e.g., recall) in class-imbalanced learning is not entirely new. For example, [a] dynamically updates reweighting costs based on the error rates in the confusion matrix (check Eq. 13). It will be appropriate to cite and discuss this paper.

- Discussing differences with some related loss functions in the imbalanced learning literature will improve Sec. 2. Examples include class rectification loss [b], cost-sensitive loss [c], range loss [d] and [e] affinity loss.

- There is a typo in Table 5, BNN should be BBN.

Refs:

[a] "Cost-sensitive learning of deep feature representations from imbalanced data." IEEE transactions on neural networks and learning systems, 2017.

[b] "Imbalanced Deep Learning by Minority Class Incremental Rectification." IEEE transactions on pattern analysis and machine intelligence, 2019.

[c] "Striking the Right Balance with Uncertainty", IEEE conference on computer vision and pattern recognition, 2019.

[d] "Range loss for deep face recognition with long-tailed training data." International conference on computer vision, 2017.

[e] “Gaussian Affinity for Max-margin Class Imbalanced Learning,” International conference on computer vision, 2019.

---

> ### Author Response · Authors · 2020-11-13
> **We reviewed suggested works by the reviewer and added citations and discussion in the revised paper.**
>
> We thank the reviewer for the insightful questions and great suggestions in terms of related work. We appreciate the reviewer’s positive comments on performance, technical analysis and potential influence of the paper.
>
> **1) Regarding decoupled structure and recall loss**\
> To find out if the recall loss improved representation learning, we included a baseline SDN(CE). The baseline is trained with the SDN architecture but using cross-entropy as the feature learning loss; this directly addresses whether the SDN architecture or our loss is responsible for the gains. Our method SDN(Recall) outperforms SDN(CE) suggesting that the proposed recall loss indeed improves feature learning. We compared our method with BBN on the iNaturalist. It is a large scale dataset with 8,412 classes. It is much larger and more complex than the other image classification dataset, Place365-LT, we tested on. Please note that the original BBN paper did not report results on the iNaturalist dataset. Furthermore, the BBN method is not directly applicable to semantic segmentation tasks.
>
> **2) Regarding the "Cost-sensitive learning" paper**\
> Thank you for this suggestion; this paper is definitely relevant to our discussion. The paper proposes to iteratively optimize both the model parameters and also a cost-sensitive layer which is integrated into the cost function. Specifically, in the proposed cost-sensitive cross entropy loss, the cost sensitive layer does not affect gradient backpropagation. It only affects the output. The loss for training the cost-sensitive layer depends on a separability matrix, a recall confusion matrix and a histogram matrix that encodes the probability of each class.  We use recall to regularize cross-entropy loss directly, resulting in a simple yet effective loss function with minimum need for tuning and faster training compared to the iterative optimization procedures in this paper. We have included this discussion in the revised paper in Appendix B and citation in the main paper.
>
> **3) Regarding related works**\
> Thank you for bringing up all of these great papers to our attention. We would like to follow up with some discussion on those and will include this discussion in the paper. While we only compared to Range loss in our experiments (table 5), the other losses are also very interesting. Most of these approaches rely on the notion of margin and clustering relationships. To enforce inter-class/intra-class constraints, [b] uses ranking loss, [c] uses uncertainty from MCDO, [d] uses harmonic mean and [e] uses multi-tasking to learn clusters. Often, these approaches have to introduce more hyperparameters([b,c,d,e]), assumptions ([c] Gaussian assumption), procedures to extract feature information ([b] example mining, [d,e] feature distance), and computational complexity ([c] MCDO). On top of added complexity, several of them have to use carefully designed training schedules to improve final performance. Our approach aims to provide an effective loss with minimum additional procedures and hyperparameters, which can be utilized to train state-of-the-art image classification and semantic segmentation networks for more efficient performance compared to other popular loss choices  (e.g., CE, Focal etc).  We have included all the citations in the main paper and discussion in Appendix B in the revised version.

---

### Author Response · Authors · 2020-11-13
**Revised paper available (changes highlighted in blue)!**

Hi,
Thanks to all the reviewers and AC for their hard work during this difficult time. We have thoroughly read through each comment and revised our paper to reflect the suggestions from the reviewers. We will have a detailed discussion of each question from our reviewers. Here we summarized the changes to the paper. We encourage the reviewers to take a look at the new additions.
1)  Added Section 3.4 on analysis of other metrics (originally in the appendix) and analytically discussed why we did not use them.
2) Added Figure 3, qualitative results on Synthia segmentation to further demonstrate the effectiveness of our method.
3) Expanded Table 5 on classification performance on Place365 and included more quantitative results for more detailed analysis
4) Updated related work and appendix to reflect all the relevant works suggested by reviewers.

In this paper, we proposed a loss function for imbalanced learning in image classification and segmentation. **We demonstrated consistently best accuracy performance on two popular *imbalance classification* benchmarks and significantly improved accuracy on two outdoor *semantic segmentation* datasets while not degrading IoU.** One of the concerns from our reviewers is that our method does not always yield the best IoU in segmentation. We would like to emphasize that our method brings significant improvement in accuracy with no/marginal drop in IoU.  This is important for safety considerations especially when the detection rate on small classes such as pedestrians, poles, bikes, can be improved by ~30%.

---

### Decision · Program_Chairs · 2021-01-07
**Final Decision**

**Decision:**

Reject

**Comment:**

This paper introduces the recall loss for dealing with imbalance training contexts. The authors propose to perform a class-wise weighting of examples based on the instantaneous recall performance during training.

The reviewers like the clarity of the presentation, but raise several concerns regarding novelty of the approach, comparison to more baseline loss functions and state-of-the-art methods.

The AC carefully reads the paper and discussions. The AC appreciates the discussion with respect to competitive loss functions, e.g. focal loss or segmentation loss approximations (SoftDice or Lovasz), which clearly highlights some limitations in existing approaches. \
However, the AC  considers that the approach is essentially a new way to setup the compromise between precision and recall. In that respect, the claims in the paper and in discussion regarding the performances of the proposed method are often exaggerated. For example in segmentation, the method obtaining the best accuracy is CB-CE in 3 out of 4 experiments (on Syntia the mIOU improvement of the recall loss compared to CB-CE corresponds to about the same drop in mACC). This is also verified on Fig 2 where CB-CE outperforms the proposed method by a large margin on mean accuracy for small classes. For classification, the performance gains' compared to SDN (CE) are small. \
The AC thus considers that the paper in the current form falls short of the ICLR acceptance threshold.